# Effect of Pretreatment on the Nitrogen Doped Activated Carbon Materials Activity towards Oxygen Reduction Reaction

**DOI:** 10.3390/ma16176005

**Published:** 2023-08-31

**Authors:** Galina Dobele, Ance Plavniece, Aleksandrs Volperts, Aivars Zhurinsh, Daina Upskuviene, Aldona Balciunaite, Vitalija Jasulaitiene, Gediminas Niaura, Martynas Talaikis, Loreta Tamasauskaite-Tamasiunaite, Eugenijus Norkus, Jannicke Kvello, Luis César Colmenares-Rausseo

**Affiliations:** 1Latvian State Institute of Wood Chemistry, Dzerbene Str. 27, LV-1006 Riga, Latvia; galina.dobele@kki.lv (G.D.); aleksandrs.volperts@kki.lv (A.V.); aivarsz@edi.lv (A.Z.); 2Center for Physical Sciences and Technology, Sauletekio Ave. 3, LT-10257 Vilnius, Lithuania; daina.upskuviene@ftmc.lt (D.U.); aldona.balciunaite@ftmc.lt (A.B.); vitalija.jasulaitiene@ftmc.lt (V.J.); gediminas.niaura@ftmc.lt (G.N.); martynas.talaikis@ftmc.lt (M.T.); loreta.tamasauskaite@ftmc.lt (L.T.-T.); 3SINTEF Industry, Batteries and Hydrogen Technologies, Strindvegen 4, NO-7465 Trondheim, Norway; jannicke.h.kvello@sintef.no (J.K.); luis.colmenares-rausseo@sintef.no (L.C.C.-R.)

**Keywords:** biomass, hybrid material, pyrolysis, hydrothermal treatment, alkali activation, mesopores, oxygen reduction reaction

## Abstract

Nitrogen-doped activated carbons with controlled micro- and mesoporosity were obtained from wood and wastes via chemical processing using pre-treatment (pyrolysis at 500 °C and hydrothermally carbonization at 250 °C) and evaluated as oxygen reduction catalysts for further application in fuel cells. The elemental and chemical composition, structure and porosity, and types of nitrogen bonds of obtained catalyst materials were studied. The catalytic activity was evaluated in an alkaline medium using the rotating disk electrode method. It was shown that an increase in the volume of mesopores in the porous structure of a carbon catalyst promotes the diffusion of reagents and the reactions proceed more efficiently. The competitiveness of the obtained carbon materials compared to Pt/C for the reaction of catalytic oxygen reduction is shown.

## 1. Introduction

The study of carbon materials from biomass and their use as electrocatalysts is of great importance, given the abundance of biomass resources on Earth, their low cost, ease of synthesis, and electrochemical properties [1]. Carbon is a key catalyst support material for oxygen reduction in proton-exchange membrane fuel cells (PEMFCs), which significantly impacts the catalyst’s structural durability and promotes the transfer of electrons to active centers, increasing the activity of the oxygen reduction reaction. Even though the whole range of necessary catalytic properties of carbon has not yet been determined, it is known that an ideal carbon support should have sufficient micro-mesoporosity, a certain degree of graphitization, and appropriate doping with specific heteroatoms [2].

For the production of activated carbons, various types of biomasses are used, including waste from its processing, such as lignins, agricultural wastes, bark, leaves, etc. The presence of carbohydrate and aromatic polymers in the biomass, as well as the use of residues of chemical processing of wood with a predominance of one or the other component, makes it possible to obtain carbon materials with different structures [3,4]. For example, it has been found that the alkaline activation of pyrolysis wood tar, which is formed after the extraction of sugars from bio-oil (levoglucosan production technology) and has high lignin content, can be used as an individual precursor or as part of a composite with wood to obtain micro-mesoporous activated carbon [5]. In the production of activated carbons preliminary carbonized precursors are usually used for processing, with thermal carbonization (or pyrolysis) being a traditional approach. However, hydrothermal carbonization (HTC) is more economically profitable, allowing to hydrolyze and transfer into an aqueous medium the least thermally stable component of wood-hemicellulose. Such carbonization products have specific properties that make them responsive precursors for further chemical activation [6,7]. Carbonaceous materials originated from HTC and traditional carbonization noticeably differ in their physicochemical properties, which affect further potential applications [8].

Activated carbons derived from biomass have been widely used for a long time, however, their application in the storage and transfer of energy (fuel cells, supercapacitors, lithium-ion batteries) has developed relatively recently [3,9]. The search for synthesis methods using various “green precursors” from biomass should become a more widespread and profitable way to meet the energy needs of society.

The use of microporous carbon materials for energy storage and conversion, despite many advantages, is complicated by an insufficiently developed ion transport structure, namely insufficient volume of mesopores and macropores which facilitate the motion of ions, and this issue requires further study. Only micropores larger than 0.5 nm can electrochemically adsorb hydrated ions and promote the formation of an electric double layer [10,11,12]. Data have been published showing that the contribution of micropores to the capacitance during fast discharge decreases in comparison with mesopores [13].

The advantage of carbon materials is the possibility of modifying them to enhance the required properties. Nitrogen-doped (N-doped) carbon materials play an increasingly important role in electrocatalysis, especially for the oxygen reduction reaction (ORR). This is due to their high efficiency in catalyzing the reaction in an alkaline medium, as well as better stability, lower cost, and broader availability compared to the commonly used expensive and depleting platinum. Heteroatoms cause significant changes in the distribution of electron density in the carbon matrix, localizing charges in the carbon atoms next to the heteroatoms. This localized charge density can act as an active site where oxygen molecules can be easily chemisorbed, effectively weakening O-O bonds and promoting its reduction [14,15,16,17]. It is important to note that the ORR activity of N-doped carbon materials depends not only on the presence of N-groups but also on the accessibility of catalytic sites due to the developed porous structure [3]. The pore size distribution and structural order of the carbon particles are among the critical parameters for assessing their potential. To create a highly efficient catalyst it is essential to find the right balance between the presence of N-groups, surface area, structure, and pore size distribution.

The electrocatalytic activity of the oxygen reduction reaction largely determines the performance of modern energy conversion and storage devices, including various fuel cells [18,19]. The method for evaluating the intrinsic activity of catalysts is based on a rotating disk electrode (RDE) and measurement of an electrode with a rotating annular disk [20]. Several valuable reviews have been published on the production of carbon electrocatalysts from biomass, including their preparation and application in ORR [21,22,23,24]. However, progress is focused on strategies for optimizing the pore structure and searching for an active site for oxygen. To evaluate the efficiency of biomass-derived carbon electrocatalysts, parameters of interest in ORR are onset potential, half-wave potential, limiting current density, and Tafel slope.

For this purpose, in this work, within the framework of the concept of waste-free use of biomass, we studied the catalytic activity of highly porous carbon materials with different mesopore volumes based on wood and liquid lignocellulose pyrolytic tar after sugars extraction (levoglucosan production waste), and their hybrid material (1:1) [5]. The resulting carbon materials with different pore size distributions were doped with nitrogen and tested as oxygen reduction catalysts for further application in fuel cells.

## 2. Materials and Methods

### 2.1. Nitrogen Doping

Activated carbons (activator, NaOH, to precursor ratio 3:1, temperature 800 °C) from wood (samples APYR-W and A-HTC-W), lignocellulose pyrolysis tar (samples APYR-T and A-HTC-T) and their mixture (1:1) (samples APYR-H and A-HTC-H) were obtained after thermal (500 °C-PYR) and hydrothermal treatment (250 °C-HTC) carbonization (synthesis and material properties described elsewhere [5]). These precursors were mixed with dicyandiamide (DCDA) solution in DMF (ratio of DCDA: precursor was 20:1). DMF was removed in a rotary evaporator and samples were treated in a muffle oven for 1 h at 800 °C in an argon atmosphere.

### 2.2. Characterization of N-Doped Carbon Materials

Carbon, nitrogen, and hydrogen content was evaluated using the Vario Macro CHNSO (Elementar Analysensysteme GmbH, Langenselbold, Germany) device. The oxygen content was calculated from the difference of 100%.

The porous structure (specific surface area, the total volume of micro- and mesopores, and the pore sizes) was determined from isotherms of low-temperature adsorption–desorption of nitrogen at 77 K on a Nova 4200e device (Quantachrome, Boynton Beach, FL, USA).

The morphology of the prepared samples was studied using transmission electron microscopy performed on the Tecnai G2 F20 X-TWIN microscope (FEI, Lincoln, NE, USA) equipped with an EDAX spectrometer with an r-TEM detector.

X-ray photoelectron spectroscopy (XPS) was used to analyze the chemical composition of the samples using a Kratos AXIS Supra+ spectrometer (Kratos Analytical Ltd, Manchester, UK) with monochromatic Al Kα (1486.6 eV) X-ray radiation powered at 225 W. The base pressure in the analysis chamber was less than 1 × 10^−8^ mbar and a low electron flood gun was used as a charge neutralizer. The survey spectra for each sample were recorded at pass energy of 80 eV with a 1 eV energy step and high-resolution spectra (pass energy–10 eV, in 0.1 eV steps) over individual element peaks. The binding energy scale was calibrated by setting the adventitious carbon peak at 284.8 eV. XPS data were converted to VAMAS format and processed using the Avantage software V5 (Thermo Scientific, East Grinstead, UK).

**Raman spectra** were recorded using an inVia Raman (Renishaw, Wotton-under-Edge, UK) spectrometer equipped with a thermoelectrically cooled (−70 °C) CCD camera and microscope. Raman spectra were excited with 532 nm radiation from diode pumped solid state (DPSS) laser (Renishaw, Wotton-under-Edge, UK). The 20×/0.40 NA objective lens and 1800 lines/mm grating were used to record the Raman spectra. The accumulation time was 40 s. To avoid damage to the sample, the laser power at the sample was restricted to 0.4 mW. The Raman frequencies were calibrated using the polystyrene standard spectrum. Parameters of the bands were determined by fitting the experimental spectra with Gaussian-Lorentzian shape components using GRAMS/AI 8.0 (Thermo Scientific, East Grinstead, UK ) software.

XRD patterns of studied powders were measured using an X-ray diffractometer D2 PHASER (Bruker, Karlsruhe, Germany) and Cu-K-alpha as an X-ray source. The measurements were conducted in the 2θ range of 10°–90°.

### 2.3. Electrochemical Measurements of the N-Doped Carbon Materials

All electrochemical measurements were performed using the Electrochemical Software Nova 2.1.6 with a Metrohm Autolab potentiostat (PGSTAT100) (Metrhom, Herisau, Switzerland). A three-electrode electrochemical cell was used for electrochemical measurements. The working electrode was a glassy carbon (GC) electrode of 5 mm diameter. The Pt sheet was used as a counter electrode and Ag/AgCl in 3M KCl as a reference electrode. The measurements were performed in the O_2_-saturated 1M KOH solution.

Before the experiments, the GC electrode was polished. The catalyst ink (2 mg mL^−1^) was obtained according to the following steps: At first, the 2 mg of the obtained catalyst was dispersed ultrasonically for 2 h in a 300 μL isopropanol, 700 μL deionized water, and 8 μL 5 wt. % Nafion. Then, 8 μL of the prepared suspension mixture was pipetted onto the GC electrode with a geometric area of 0.19625 cm^2^ and dried at room temperature overnight. The catalyst loading was 88.9 μg cm^−2^.

Linear-sweep voltammograms (LSVs) were recorded with a scan rate of 10 mV s^−1^ at rotation rates (ω) from 100 to 2400 rpm in an O_2_-saturated 1 M KOH solution. The data were collected at 100, 400, 800, 1200, 1600, 2000, and 2400 rpm. The electrode potential is quoted versus the reversible hydrogen electrode (RHE). The presented current densities are normalized to the geometric area of catalysts.

The accelerated durability test for samples was conducted by cycling the samples 1000 times between 1.0 to 0.6 V vs. RHE at 800 rpm with a scan rate of 200 mV s^−1^. After 1000 cycles an O_2_ LSV was recorded from 1.0 to 0.25 V vs. RHE at a scan rate of 10 mV s^−1^.

The Ossila four-point probe system (Ossila Ltd., Sheffield, UK) was used to measure the sheet resistance of the material. Sheet resistance is the resistance of the material divided by its thickness and represents the lateral resistance through a thin square of conductive material. For this measurement, four probes were used, arranged in a line with equal space between each one. Current is passed between the outer probes, causing a voltage drop between the two inner probes. By measuring this voltage change, the resistance of the sheet can be calculated.

## 3. Results and Discussion

### 3.1. Characterisation of N-Doped Carbon Materials

In this study, textural and electrocatalytic properties of activated carbon materials obtained from wood, water-insoluble lignocellulose pyrolysis tar, and their mixture (1:1) were studied. The samples were carbonized thermally (pyrolysis 500 °C) and hydrothermally (250 °C) and then chemically activated with NaOH (alkali to precursor ratio 3:1, temperature 800 °C) and demineralized. In previous works, we have shown the effectiveness of using the cheaper alkali NaOH compared to widespread KOH as a chemical activator of various biomass samples for the formation of a nanoporous structure of carbon materials [25].

The samples obtained after activation, despite the different chemical composition and carbonization conditions, only slightly differed in elemental composition: the carbon content was from 91.7 to 93.4% and oxygen from 4.5 to 6.4% (Table 1) and the developed specific porous surface, which is typical for microporous materials, was more than 2300 m^2^ g^−1^ (Table 1, Figure 1a). The main effect of different carbonization conditions was manifested in the overall increase in pore volume and the proportion of mesopores in the case of hydrothermal pretreatment, compared to those of thermal carbonization (Table 1, Figure 1b).

Activation of precursors obtained hydrothermally leads to high volumes of pores with sizes of 2.5–4.5 nm (mesopores) (Table 1, Figure 1b). The volume of the pores wider than 2 nm in the total pore volume is 55–59%, while the mesopore volume for thermally carbonized precursors is only 41–50%.

After doping with nitrogen using dicyandiamide as a dopant, the porous structure practically does not change since it was formed during activation; however, as reported in many studies [26], the presence of a nitrogen heteroatom has a significant effect on reduction reactions, increasing catalytic activity.

It has been reported [27] that the nitrogen heteroatom functionalities affect the electrochemical properties. Elemental analysis and X-ray photoelectron spectroscopy (XPS) were performed to evaluate the elemental compositions, and the results were compiled in Table 1 and Figure 2. Both elemental analysis and XPS overview (Figure 2a,b) spectra confirmed the presence of carbon, nitrogen, and oxygen in all samples. The deconvoluted high-resolution spectra of the N1s peak of all samples are shown in Figure 2c,d. All samples have a predominant pyridinic-N form at 398.2 eV binding energy (APYR-W-N—48%, APYR-H-N—58%, APYR-T-N—62%, AHTC-W-N—52%, AHTC-H-N—53%, AHTC-T-N—50%). Experiments [28,29] and theoretical calculations described in the literature have shown that nitrogen in this form is the most active in the oxygen reduction reaction. The second most represented form is pyrrolic-N at 400.6 eV (APYR-W-N—48%, APYR-H-N—58%, APYR-T-N—62%, AHTC-W-N—52%, AHTC-H-N—53%, AHTC-T-N—50%). APYR-W-N (16%), AHTC-H-N (7%), and AHTC-T-N (31%) have graphitic-N form at 402.4 eV. Lai et al. [30] found that graphitic N increases the limiting current density and that pyridinic N improves onset potential. Only AHTC-W-N has N-O at 405 eV. One of the most important factors influencing the relative percentage of N species in carbon materials is the treatment temperature, which has a profound effect on both the quantity of N content and nitrogen chemistry [31]. It has been reported, that at temperatures below 800 °C more pyridinic N was observed in the structure of carbonaceous material [32]. Further elevation of treatment temperatures can lead to the partial destruction of the structural order of the samples [31].

Several C chemical states were confirmed on the surface of carbons by the high resolution of C1s peak (Figure 2e,f), which can be deconvoluted into peaks at around 284.7, 285.3, and 286.5 eV, corresponding to C–C/C=C, C–O/C=N, and C=O/C–N and O–C=O/O=C–N, respectively [33]. Most of the carbon atoms remain bonded together in a conjugated honeycomb pattern (sp2-C and sp3-C bonds). The main component in all obtained samples corresponds to the presence of C atoms as graphitelike sp2 C-C bonds, especially in the case of AHTC-N.

Resonance Raman spectroscopy was applied to access the structural distortion and defects in N-doped activated carbon materials. Figure 3 shows Raman spectra of differently prepared carbon samples. All the samples exhibited two strong bands near 1343–1348 and 1582–1606 cm^−1^ assigned to D and G vibrational modes, respectively.

The Raman-allowed G-mode belongs to E_2g_ symmetry and is associated with the in-plane relative motion of pairs of sp^2^-hybridized carbon atoms [34,35]. The D-mode belongs to A_1g_ symmetry and can be described as the breathing vibration of six-membered aromatic carbon rings. This mode is forbidden in idealized graphene structures and appears due to the presence of defects and disorders [34,35]. Some samples exhibit higher frequency features in the vicinity of 2685–2694 cm^−1^ assigned to the overtone of D-band (2D mode). Various parameters derived from Raman spectra are listed in Table 2. The degree of disorder associated with point-like defects due to the destruction of sp^2^-hybridization can be evaluated by analysis of the intensity ratio of D and G bands, I(D)/I(G) [36,37]. Relatively low I(D)/I(G) values were found for APYR-W-N and APYR-H-N samples, indicating lower disordering and higher structural quality of these carbon materials (Table 2). Other studied samples exhibited similar I(D)/I(G) ratios. The width of the G band determined as the full width at half maximum intensity [FWHM(G)] can be employed as a measure of graphitization of (transformation of less ordered structures to highly ordered forms) carbon samples [38]; higher widths reflect a lower degree of graphitization. Both parameters, I(D)/I(G) and FWHM(G), increase as crystallite size decreases.

However, the intensity ratio is better suitable for the analysis of samples with crystallite sizes higher than 30 nm, while FWHM(G) probes the organization of lamellas with smaller crystalline sizes (down to 3 nm) [39]. Again, both APYR-W-N and APYR-H-N samples showed relatively small FWHM(G) values, while the APYR-T-N material expressed the highest width of the G band (Table 2).

The samples ANTC-W-N, ANTC-H-N, and ANTC-T-N also show broad G bands consistent with low graphitization degree unordered structure. The third important Raman parameter of carbon material is the integrated intensity of the so-called D’’ band in the vicinity of 1490–1530 cm^−1^ [34,40,41,42]. This band is not clearly visible but is responsible for increased background in the vicinity between the D and G modes. The parameters of this band were evaluated by the fitting of the experimental contour with Lorentzian-Gaussian form components [42]. Integrated intensity of the D’’ band [A(D’’)] can serve as an indicator of the amorphization of carbon material [34,40,41,42]. One can see that the A(D’’) value considerably increases in the case of samples APYR-T-N, AHTC-W-N, AHTC-H-N, and AHTC-T-N; in particular, the samples APYR-T-N and AHTC-T-N exhibit highest integrated intensity values. Thus, these materials contain a high amount of amorphous-like carbon structure.

Only two studied samples demonstrated the high intensity of the 2D band. It was revealed that the intensity ratio I(2D)/I(G) provides the possibility to identify the number of graphene layers [35,43,44]. In the case of a single graphene layer, the I(2D)/I(G) > 1, and the form of a 2D band should correspond to a single Lorentzian component with a width FWHM(2D) of ~24 cm^−1^ [44]. It was demonstrated that the FWHM(2D) value could serve as a measure of layer number in few-layer graphene samples [45]. In the 532 nm excited Raman spectra, the FWHM(2D) increased from 26.3 cm^−1^ in the case of single-layer graphene to 52.1, 56.1, and 62.4 cm^−1^ for two-, tri-, and four-layer graphene structures, respectively. Analysis of our samples APYR-W-N and APYR-H-N revealed the I(2D)/I(G) ratios of 0.48 and 0.66, respectively; the FWHM(2D) values were found to be 61.5 and 61.3 cm^−1^, respectively. Thus, these two samples can be considered as four-layer graphene structures.

The XRD results shown in Figure 4. confirm the Raman results, proving that materials that have been pyrolyzed before activation (APYR-W-N, APYR-H-N and APYR-T-N) have a more ordered crystalline structure and peaks can be observed at 26.5°, 43°, 44° and 64°, which correspond to C (002), C (100), C (101) (orientated aromatic structure, crystal planes of graphitic carbon [46,47]) and (103) (crystal plane reflections assigned to graphite [48]). On the other hand, no peaks have been observed for hydrothermally carbonized samples, their structure is completely amorphous. It should be noted that while all obtained samples are generally amorphous, only pyrolyzed samples have crystallites of various sizes and orientations as a part of the amorphous carbon matrix. Peak observed at 37.5° (311), 43.3 (400) and 62.9 (440) can be related to magnetite (Fe_3_O_4_) in the structure of the studied materials [49,50]. Metal impurities may appear in the pyrolysis process where iron bound to the carbon matrix can further influence, even enhance, the catalytic activity [51].

To obtain more information on the texture characters of the synthesized carbon matrices, TEM was performed (Figure 5). TEM images show that all samples have heterogeneous textures, with areas of crumpled and wavy sheets, multilayer graphene-like structures, and denser amorphous structures.

The crumpled structure increases the surface area of the nanosheets and reduces the process of stacking their interlayers π–π bonds, thus promoting the formation of porous layers [52]. A distinctly amorphous structure can be observed in the case of hydrothermally carbonized and pyrolysis tar, while the more ordered structure is formed in the process of carbonized wood.

### 3.2. Electrochemical Tests

The ORR performance was evaluated using the rotating disk electrode (RDE) method. ORR polarization curves were recorded in the 1 M KOH solution at a scan rate of 10 mV s^−1^ over a range of rotation rates from 100 to 2400 rpm. The data of RDE measurements at different rotation rates for pyrolyzed samples (APYR-W-N, APYR-H-N, and APYR-T-N), hydrothermally carbonized samples (AHTC-W-N, AHTC-H-N, and AHTC-T-N), and Pt/C are given in Appendix A.

A comparison of ORR LSVs recorded for the pyrolyzed samples (APYR-W-N, APYR-H-N, and APYR-T-N), hydrothermally carbonized (AHTC-W-N, AHTC-H-N, and AHTC-T-N) and commercial Pt/C catalyst (Tanaka, 46.4 wt.% Pt) in O_2_-saturated 1M KOH solution at the 1600 and 2200 rpm are presented in Figure 6a and Figure 6b, respectively. The data of onset potential (*E*_onset_), half-wave potential (*E*_1/2_), and the average number of electrons transferred per O_2_ molecule (*n*) are presented in Table 3. As evident, the studied samples are active ORR catalysts in alkaline media, showing an onset potential of 0.90–0.92 V and a corresponding half-wave potential of 0.81–0.84 V, which are close to the *E*_onset_ and *E*_1/2_ values for the commercial Pt/C catalyst, i.e., 0.95 V and 0.88 V, respectively (Figure 6a,b and Table 3).

A comparison of the diffusion limiting current density of the two subsets of the samples under study (namely APYR and AHTC) shows that with the increase in the rotation speed in the case of APYR samples, limiting current does not change as significantly as it can be overserved in the case of AHTC samples.

This is especially pronounced for the AHTC-T-N sample (Figure 6b), with the limiting current being close to that of the Pt/C catalyst. Judging by the pore size distribution, the AHTC-T-N sample has the highest mesoporous volume of all the materials under study (Table 1, Figure 1a), and possibly, this feature of the structure under conditions of an increase in the rotation speed accelerates the diffusion of the reagent and activates diffusion in a larger volume of pores. These results indicate higher reactivity due to the presence of reaction sites for AHTC samples, highlighting the importance of the contribution of mesopores to the total pore volume of the material structure (Table 1).

The Tafel analysis was used to evaluate the ORR activity and quality of the synthesized N-doped carbon materials. The empirical Tafel equation (Equation (1)) could give a linear region between the logarithms of measured current density and the applied potential:η = a + b × log*j*(1)
where η (V) is the applied overpotential, a (V) is the curve intercept, b (V dec^−1^) is the Tafel plot, and *j* (A cm^−2^) is the resulting current density. Tafel slope (c,d) indicates that the current density changes as a function of rising overpotential. A smaller Tafel slope means that the low overpotential needs to reach a high current density. Figure 6c,d presents the mass-transfer corrected Tafel plots using the Koutecky-Levich equation, where the kinetic current density for ORR (*j*_k_) could be calculated according to Equation (2):*j*_k_ = (*j*_L_ × *j*)/(*j*_L_ − *j*)(2)
where *j*_L_ is the O_2_-diffusion-limited current density at 0.25 V vs. RHE measured at 1600 and 2200 rpm, and *j* is the Faradaic current density. The low Tafel slope values (−49.40 ÷ −62.30 mV dec^−1^) were obtained at 1600 rpm and 2200 rpm for APYR-W-N, APYR-H-N, APYR-T-N, AHTC-W-N, AHTC-H-N, and AHTC-T-N, indicating high activity for ORR (Figure 5c,d, Table 3). Consequently, AHTC-T-N shows the largest limiting current density at 0.25 V and the smallest Tafel slope of 49.40 mV dec^−1^ among the studied APYR and AHTC samples. Notably, the AHTC-T-N catalyst with the predominance of mesopores exhibited the highest activity towards ORR and comparable activity to the commercial Pt/C (Tanaka) catalyst in terms of limiting current density and a low Tafel slope.

The number of electrons transferred per O_2_ molecule (n) was calculated using Koutecky–Levich (K-L) Equations (3)–(5) [53]:1/*j* = 1/*j*_k_ + 1/*j*_d_ = 1/B*ω*^1/2^ + 1/*j*_k_
(3)
B = 0.62nFC_0_(D_0_)^2/3^ν^−1/6^
(4)
*j_k_* = nFkC_0_
(5)
where *j*, *j*_k_, and *j*_d_ are the experimentally measured current density, kinetic and diffusion-limiting current densities, respectively; k is the electrochemical rate constant for O_2_ reduction, C_0_ is the concentration of oxygen in the bulk (7.8 × 10^−7^ mol cm^−3^), F is the Faraday constant (96,485 C mol^−1^), D_0_ is the diffusion coefficient of O_2_ (1.8 × 10^−5^ cm^2^ s^−1^), ν is the kinematic viscosity of the solution (0.01 cm^2^ s^−1^), and ω is the rotation rate of the electrode (rad s^−1^).

The number of electrons transferred per O_2_ molecule was calculated from the Koutecky–Levich (K-L) plots (see Appendix A) and is presented in Table 3 and Figure 7. The calculated *n* values for studied catalysts can be divided into two groups, those which are close to 4 or reaching over 4 in the potential range from 0.25 to 0.65 V, meaning the direct 4e^−^ reduction pathway from O_2_ to H_2_O observed for all pyrolyzed, activated and nitrogen-doped catalysts and those which are higher than 3 with a combined 2 e^−^ and 4 e^−^ mechanisms, observed for hydrothermally carbonized, activated and nitrogen-doped carbons. The obtained higher *n* values than 4 for APYR-W-N and APYR-T-N indicate that the catalyst layers on the GC electrode were rather porous and rough [54].

Thus, a comparison of the catalytic activity of pyrolyzed and hydrothermally carbonized activated and nitrogen-doped carbons based on various biomass samples showed that they could be competitive for Pt/C at ORR. At the same time, hydrothermal carbonization has a positive effect on the formation of mesopores, increasing the diffusion of reagents and the catalytic activity of samples.

According to the XRD data, samples that have been pyrolyzed before activation, rather than hydrothermally carbonized, have some iron impurities and it is possible that it could influence the catalytic activity of these samples [55].

Accelerated durability tests for all catalysts were conducted by cycling the catalyst 1000 times between 1.0 to 0.6 V vs. RHE at 800 rpm with a scan rate of 200 mV s^−1^. After 1000 cycles, an O_2_ LSV was recorded from 1.0 to 0.25 V vs. RHE at a scan rate of 10 mV s^−1^ (Figure 8). As evident in Figure 8, among investigated catalysts, no noticeable changes in onset potential and half-wave potential can be observed for AHTC-W-N (d), when comparing ORR activities of this sample before and after 1000 potential cycles, indicating a high stability of this catalyst. Lower negative shifts are also observed in the *E*_1/2_ potential for other hydrothermally carbonized samples (AHTC-H-N and AHTC-T-N) (ca. 10.7 and 14.1 mV) (Figure 8e,f, Table 4) as compared with the pyrolyzed samples (APYR-W-N, APYR-H-N, and APYR-T-N) (ca. 13–28 mV) (Figure 8a–c, Table 4). In regard to the Pt/C catalyst (Figure 8g), an obvious negative shift (25.2 mV) is observed in the onset potential and half-wave potential. The stability tests show that AHTC-W-N, AHTC-H-N, APYR-W-N, and AHTC-T-N samples have excellent stability as compared to the commercial Pt/C catalyst. APYR-H-N and APYR-T-N samples exhibit similar stability to the obtained stability test of Pt/C.

Notably, pyrolyzed and hydrothermally carbonized nitrogen-doped carbons reached a high electrical conductivity of 3.48–4.22 × 10^6^ S cm^−1^, which can be associated with electronic doping induced by substitutional nitrogen atoms [56]. Notably, the dependence of the conductivity of pyrolyzed and hydrothermally carbonized nitrogen-doped carbons on the nitrogen content is non-monotone. The highest conductivity was obtained for APYR-T-N catalyst (4.22 × 10^6^ S/cm) with the concentration of nitrogen 3.46 at.% (Table 5).

## 4. Conclusions

Pyrolyzed (500 °C) and hydrothermally carbonized (250 °C), then alkali-activated and nitrogen-doped samples from wood and waste from its chemical processing, as well as their hybrid composite, were tested as oxygen reduction catalysts. All obtained samples had high specific surface areas exceeding 2300 m^2^ g^−1^ and differed in mesopore volumes. Hydrothermal carbonization of materials, in contrast to thermal carbonization, contributed to the prevalence of the mesoporous structure. The XPS method was used to study the types of bonds of the introduced nitrogen; the results showed that most electrochemically active pyridine form is predominant (more than 47% of the total N) for all samples. According to the Raman spectra, the pyrolytically carbonized samples APYR-W-N and APYR-H-N can be considered as four-layer graphene structures (ratio I(2D)/I(G) = 0.48 and 0.66, respectively, and full width at half maximum of 2D band, FWHM(2D) = 61.5 and 61.3 cm^−1^, respectively). Relatively low I(D)/I(G) values were found for other samples, indicating less disorder and higher structural quality of these carbon materials.

All obtained catalyst materials exhibit high catalytic activity for ORR in an alkaline medium, showing an onset potential of 0.90–0.92 V and a corresponding half-wave potential of 0.81–0.84, and low Tafel slopes (49–62 mV dec^−1^), which are close to those for the commercial Pt/C catalyst. The catalytic reducing ability can be explained by a high specific surface, micro-, mesoporous structure, and activity of nitrogen catalytic sites. The origin of activated biomass samples plays a minor role in ORR activity, apparently due to the preserved hierarchical structure. It is shown that the predominance of mesopores in the hydrothermally carbonized samples increases the diffusion of reagents and the reduction activity of the samples, contributing to the limiting current density and catalytic activity of the samples. Additionally, hydrothermally carbonized samples have excellent stability compared to the commercial Pt/C catalyst and are potential candidates to replace platinum in low-temperature fuel cells.

## Figures and Tables

**Figure 1 materials-16-06005-f001:**
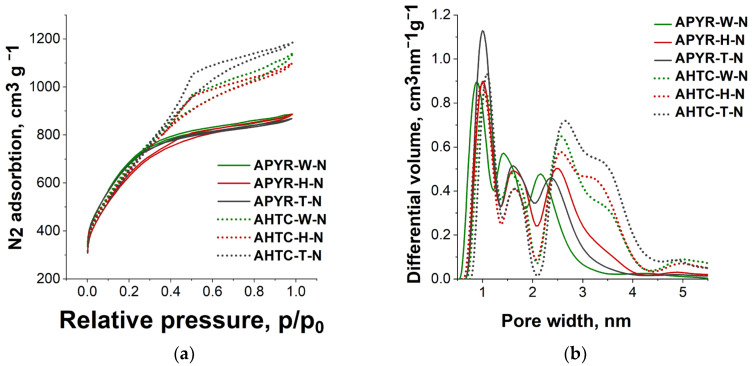
(**a**) N_2_ adsorption–desorption isotherms at 77 K and (**b**) pore size distribution of N-doped activated carbons based on hydrothermally carbonized samples (AHTC-W-N, AHTC-H-N, and AHTC-T-N) and pyrolyzed samples (APYR-W-N, APYR-H-N, and APYR-T-N).

**Figure 2 materials-16-06005-f002:**
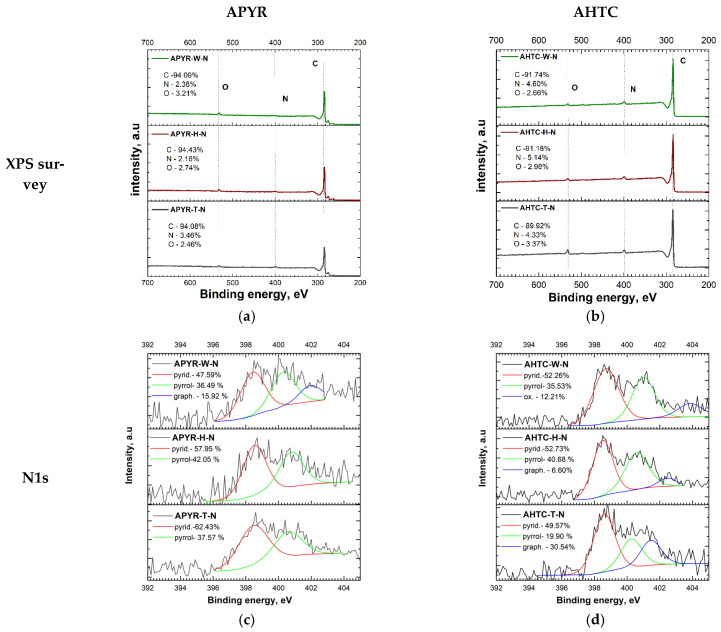
(**a**,**b**) XPS survey, (**c**,**d**) N 1s spectra and (**e,f**) C1s of N-doped activated carbon on activated carbon based on hydrothermally carbonized samples (AHTC-W-N, AHTC-H-N, and AHTC-T-N) and pyrolyzed samples (APYR-W-N, APYR-H-N, and APYR-T-N).

**Figure 3 materials-16-06005-f003:**
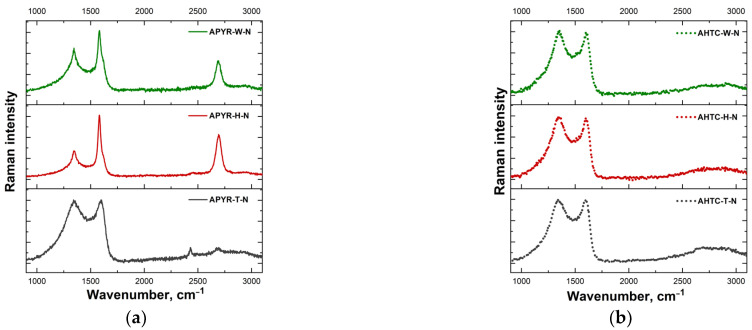
Raman spectra of N-doped carbon-based samples. (**a**) APYR-W-N, APYR-H-N, and APYR-T-N and (**b**) AHTC-W-N, AHTC-H-N, and AHTC-T-N. The excitation wavelength is 532 nm (0.4 mW).

**Figure 4 materials-16-06005-f004:**
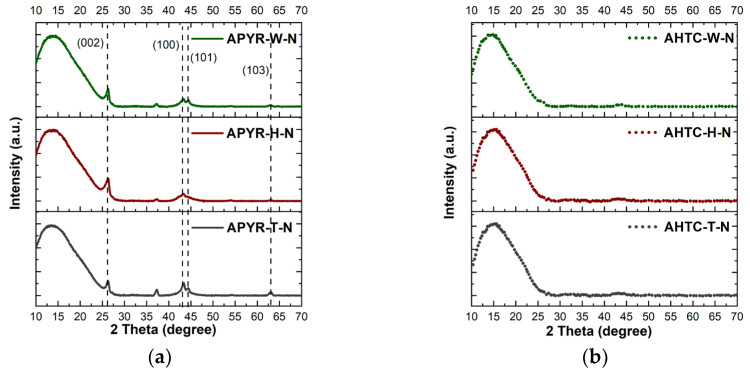
XRD study of (**a**) APYR-W-N, APYR-H-N, and APYR-T-N and (**b**) AHTC-W-N, AHTC-H-N, and AHTC-T-N.

**Figure 5 materials-16-06005-f005:**
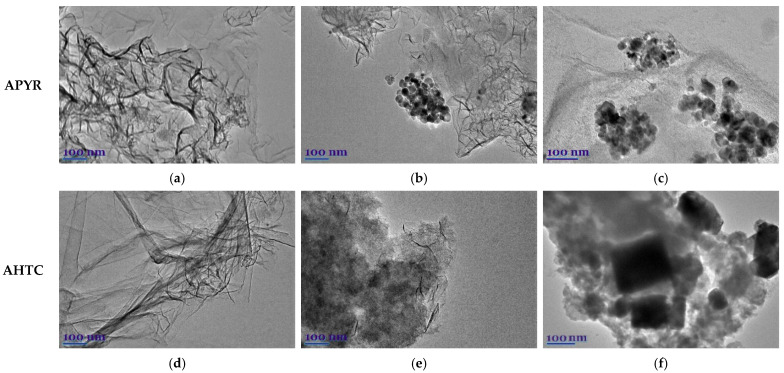
TEM images of N-doped activated carbon based on pyrolyzed samples (APYR-W-N (**a**), APYR-H-N (**b**) and-APYR-T-N (**c**)) and after hydrothermal carbonization (AHTC-W-N (**d**), AHTC-H-N (**e**), and AHTC-T-N (**f**)).

**Figure 6 materials-16-06005-f006:**
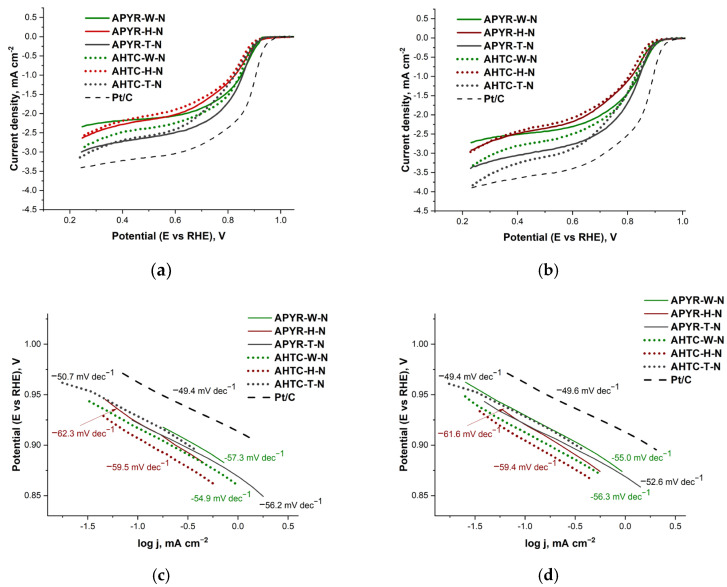
LSVs recorded on pyrolyzed samples (APYR-W-N, APYR-H-N, APYR-T-N), samples after hydrothermal carbonization (AHTC-W-N, AHTC-H-N, and AHTC-T-N) and Pt/C catalyst at a scan rate of 10 mV s^−1^ in O_2_-saturated 1 M KOH at the rotation rate of 1600 rpm (**a**) and 2200 rpm (**b**). Tafel plots derived from LSV curves at the rotation rate of 1600 rpm (**c**) and 2200 rpm (**d**).

**Figure 7 materials-16-06005-f007:**
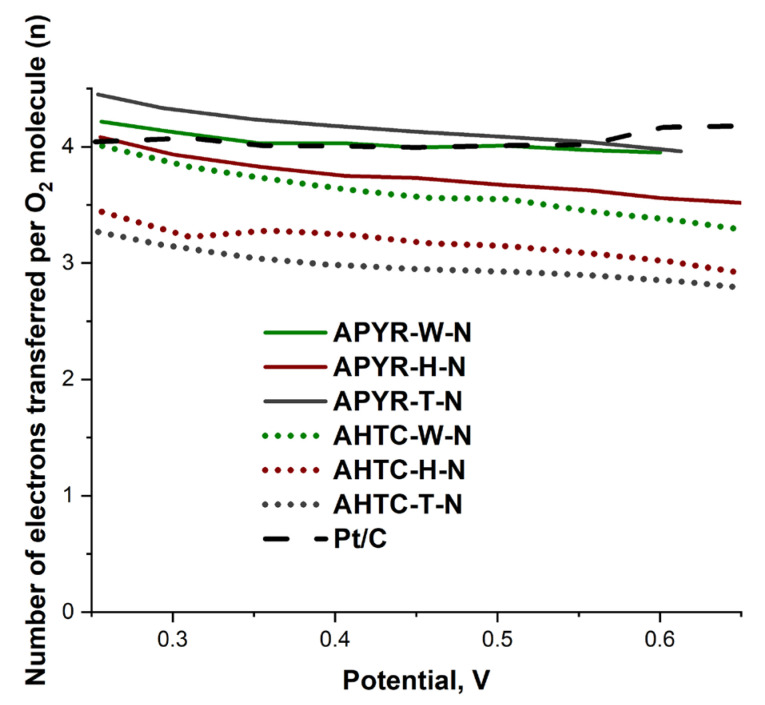
The potential dependence of the number of electrons transferred per O_2_ molecule (*n*).

**Figure 8 materials-16-06005-f008:**
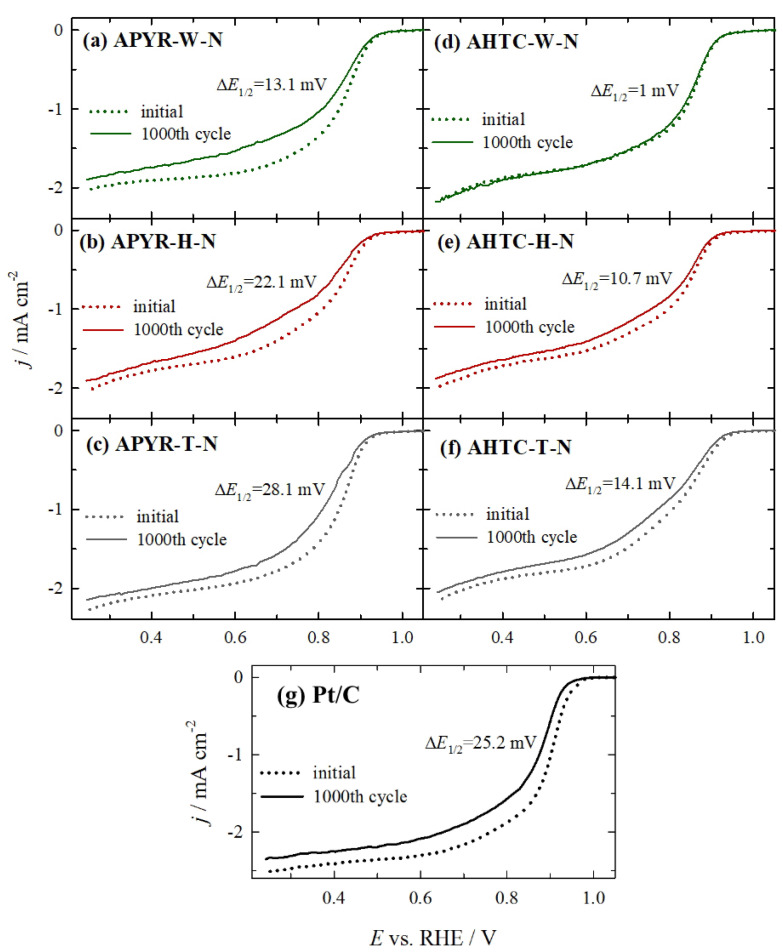
The stability results including the ORR polarization curves of the APYR-W-N (**a**), APYR-H-N (**b**), APYR-T-N (**c**), AHTC-W-N (**d**), AHTC-H-N (**e**), AHTC-T-N (**f**) and Pt/C (**g**) catalysts before and after 1000 potential cycles in O_2_-saturated 1 M KOH at 10 mV s^−1^. Rotation rate 800 rpm.

**Table 1 materials-16-06005-t001:** Porous structure parameters and elemental composition of activated carbon based on hydrothermally carbonized samples (AHTC-W, AHTC-H, and AHTC-T) and pyrolyzed samples (APYR-W, APYR-H, and APYR-T) before and after N-doping.

Sample	Yield, %	S_BET_, m^2^ g^−1^	V_total_, cm^3^ g^−1^	V_meso_ from V_t_, %	N, %	C, %	H, %	O, %
APYR-W	10	2553	1.4	41	0.9	92.0	0.6	6.4
APYR-H	14	2494	1.4	44	0.7	92.5	0.7	6.1
APYR-T	14	2804	1.5	40	0.8	93.0	0.7	5.5
AHTC-W	9	2629	1.9	58	1.7	91.7	0.6	6.0
AHTC-H	7	2919	2.2	59	1.2	92.0	0.6	6.1
AHTC-T	7	2401	1.9	58	0.6	93.4	1.3	4.5
APYR-W-N	9	2497	1.4	44	3.4	91.0	2.2	3.3
APYR-H-N	14	2306	1.4	49	3.5	89.8	2.5	4.2
APYR-T-N	14	2482	1.3	42	5.0	89.6	1.8	3.6
AHTC-W-N	8	2431	1.6	57	5.1	91.8	0.9	2.3
AHTC-H-N	7	2395	1.7	55	5.3	91.3	0.6	2.7
AHTC-T-N	7	2357	1.8	59	4.0	91.4	0.4	4.2

**Table 2 materials-16-06005-t002:** Relative intensity of D and G Raman bands [I(D)/I(G)], full width at half maximum of G band [FWHM(G)], and relative integrated intensity of D″ band [A(D″)].

Sample	I(D)/I(G)	FWHM(G) (cm^−1^)	A(D″)
APYR-W-N	0.70	39.7	132
APYR-H-N	0.42	31.8	100
APYR-T-N	1.00	74.6	409
AHTC-W-N	1.05	69.7	357
AHTC-H-N	1.04	65.6	357
AHTC-T-N	1.02	71.3	391

**Table 3 materials-16-06005-t003:** The data of onset potential, half-wave potential, the average number of electrons transferred per O_2_ molecule (*n*), and Tafel plots.

		At 1600 rpm			At 2200 rpm			
Sample	*E*_onset_, V	*E*_1/2_, V	Tafel Plot, mV dec^−1^	*j*_L_ at 0.25 V	*E*_1/2_, V	Tafel Plot, mV dec^−1^	*j*_L_ at 0.25 V	Average *n*
APYR-W-N	0.91	0.84	−57.30	−2.32	0.84	−55.00	−2.70	4.0
APYR-H-N	0.91	0.82	−62.30	−2.60	0.81	−61.60	−2.90	3.8
APYR-T-N	0.91	0.84	−56.20	−2.97	0.84	−52.60	−3.07	4.2
AHTC-W-N	0.92	0.84	−54.90	−2.89	0.83	−55.50	−3.33	3.6
AHTC-H-N	0.90	0.82	−59.50	−2.58	0.82	−59.20	−2.94	3.1
AHTC-T-N	0.92	0.81	−50.70	−3.14	0.81	−49.40	−3.82	3.0
Pt/C	0.95	0.89	−49.40	−3.39	0.89	−49.60	−3.89	4.0

**Table 4 materials-16-06005-t004:** The data of half-wave potential before and after 1000 potential cycles. Data were collected at 800 rpm.

Sample	E_1/2_, V, Initial	E_1/2_, V, after 1000th Cycles	ΔE, mV
APYR-W-N	0.856	0.843	13.1
APYR-H-N	0.843	0.821	22.1
APYR-T-N	0.849	0.821	28.1
AHTC-W-N	0.851	0.850	1.0
AHTC-H-N	0.837	0.826	10.7
AHTC-T-N	0.835	0.821	14.1
Pt/C	0.897	0.872	25.2

**Table 5 materials-16-06005-t005:** Conductivity and resistivity of investigated carbon samples.

Sample	Conductivity, × 10^6^ S/cm	Resistivity, × 10^7^ Ohm·cm	N, at.%
APYR-W-N	3.48	2.87	2.38
APYR-H-N	3.85	2.60	2.16
APYR-T-N	4.22	2.37	3.46
AHTC-W-N	4.18	2.39	4.60
AHTC-H-N	3.63	2.75	5.14
AHTC-T-N	3.60	2.77	4.33

## Data Availability

The data presented in this study are available on request from the corresponding author.

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
