# Peer review of "Effect of Pretreatment on the Nitrogen Doped Activated Carbon Materials Activity towards Oxygen Reduction Reaction"

_materials, 2023, doi:10.3390/ma16176005_

Round 1

Reviewer 1 Report

The article «Effect of Pretreatment On The Nitrogen Doped Activated Carbon Materials Activity Towards Oxygen Reduction Reaction» is devoted to the production of new high-performance nitrogen-doped carbon supports used as catalysts for the oxygen reduction reaction in an alkaline medium. The authors use a set of modern methods for studying the structure of the obtained N-doped materials and study in detail their catalytic activity on RDE. The article is well written, the results obtained are of undoubted interest. The only caveat is the need to use RRDE measurements to measure the amount of H2O2 released on various materials.

Author Response

Thank you for the suggestion, RRDE measurement are planned for the next stage of our research and will be published in the follow-up paper.

Reviewer 2 Report

The authors use pre-treatment (pyrolysis (500°Ð¡) 12 and hydrothermally carbonization (250°Ð¡)) to study the characteristics of Nitrogen-doped activated carbons. The elemental and chemical composition, structure and porosity, and types of nitrogen bonds of 14 obtained catalyst materials have been studied. This work is essential for the electrochemical applications. However, the following questions should be answered before publication.  (1) In Fig.2, the author discussed the form of N in experiments, and stated the opinion from literature on the difference of N form. It is suggested to discuss the effect of nitrogen doping condition on the composition of different N forms.  (2) In Fig.4, the author showed the different structure, such as crumpled and wavy sheets, a multilayer graphene-like structure and a denser amorphous structure. A literature is offered to discuss the effect of crumpled structure, while other structures are not discussed.  (3) In Fig.7, the author claimed that AHTC-W-N has no changes after 1000 cycle. What is the reason for this stability? Is it due to the structure? Besides, it is shown in Fig.7(d) that after 1000 cycle the j is even smaller than the original value, why?

Author Response

The authors use pre-treatment (pyrolysis (500°Ð¡) 12 and hydrothermally carbonization (250°Ð¡)) to study the characteristics of Nitrogen-doped activated carbons. The elemental and chemical composition, structure and porosity, and types of nitrogen bonds of 14 obtained catalyst materials have been studied. This work is essential for the electrochemical applications. However, the following questions should be answered before publication. 

(1) In Fig.2, the author discussed the form of N in experiments, and stated the opinion from literature on the difference of N form. It is suggested to discuss the effect of nitrogen doping condition on the composition of different N forms. 

Short discussion and references were added into the manuscript.

 (2) In Fig.4, the author showed the different structure, such as crumpled and wavy sheets, a multilayer graphene-like structure and a denser amorphous structure. A literature is offered to discuss the effect of crumpled structure, while other structures are not discussed. 

Thank you for the suggestion.

since crumpled and wavy sheets, a multilayer graphene-like structure has the largest contribution, it is described in more details.

 (3) In Fig.7, the author claimed that AHTC-W-N has no changes after 1000 cycle. What is the reason for this stability? Is it due to the structure? Besides, it is shown in Fig.7(d) that after 1000 cycle the j is even smaller than the original value, why?

The reason for AHTC-W-N stability may be related to the catalyst morphology. The high stability of this catalyst indicates that during the stability test, the catalyst morphology on the electrode does not changes, and the catalyst material is not detached from the electrode. The original j value and the one after the 1000 cycle test differ because the stability test was carried out at a lower rpm value of 800 rpm, whereas the ORR was investigated at a higher rpm value of 1600 rpm.

Reviewer 3 Report

Manuscript Number: materials-2427639

Title: Effect of Pretreatment On The Nitrogen Doped Activated Carbon Materials Activity Towards Oxygen Reduction Reaction.

In this manuscript, activated carbons doped with nitrogen were prepared from wood and its chemical processing wastes through pre-treatment techniques, and their potential as catalysts for oxygen reduction in fuel cells was examined. The catalytic activity was assessed under alkaline conditions using the rotating disk electrode method. The results demonstrated that an increased volume of mesopores in the carbon catalyst enhanced reagent diffusion and improved reaction efficiency. Because the manuscript is well organized, the investigation is comprehensive, and the results are significant, I consider it suitable for publication in Materials after the following problems are solved.

1. The authors state, "The competitiveness of the obtained carbon materials compared to Pt/C for the reaction of catalytic oxygen reduction is shown," and "This is especially pronounced for the AHTC-T-N sample (Fig. 5 b), with the limiting current being close to that of the Pt/C catalyst." Until now, platinum has been considered the optimal catalyst for the oxygen reduction reaction. Non-Pt catalyst materials have struggled to compete with Pt/C catalysts for the oxygen reduction reaction. Therefore, the authors need to further demonstrate why Nitrogen-doped activated carbons have the ability to provide such a good diffusion limiting current density value.

2. The overall quality of the graphics is below the general standards (e.g., Figure 1, Figure 2, Figure 5, Figure 7, …);  improvements in style, clarity, contrast, and sharpness are needed.

3. The authors should provide XRD and TEM images for Pt/C.

4. Electronic conductivity plays an important role in the electrocatalyst materials. The authors need to provide the conductivity of the synthesized material samples.

5. Some references such as https://doi.org/10.1039/D0CY02056G; https://doi.org/10.1039/D2DT01268E  should be mentioned when discussing the Oxygen Reduction Reaction.

Author Response

  1. The authors state, "The competitiveness of the obtained carbon materials compared to Pt/C for the reaction of catalytic oxygen reduction is shown," and "This is especially pronounced for the AHTC-T-N sample (Fig. 5 b), with the limiting current being close to that of the Pt/C catalyst." Until now, platinum has been considered the optimal catalyst for the oxygen reduction reaction. Non-Pt catalyst materials have struggled to compete with Pt/C catalysts for the oxygen reduction reaction. Therefore, the authors need to further demonstrate why Nitrogen-doped activated carbons have the ability to provide such a good diffusion limiting current density value.

The paper was improved to reflect the properties of N-doped carbons.

  1. The overall quality of the graphics is below the general standards (e.g., Figure 1, Figure 2, Figure 5, Figure 7, …);  improvements in style, clarity, contrast, and sharpness are needed.

The pictures quality was improved.

  1. The authors should provide XRD and TEM images for Pt/C.

Since it is widely known and used catalyst the information on its structure and properties can be found in numerous papers, for example:

AyÅŸe Bayrakçeken, Alevtina Smirnova, Usanee Kitkamthorn, Mark Aindow, Lemi Türker, Ä°nci EroÄŸlu, Can Erkey, Pt-based electrocatalysts for polymer electrolyte membrane fuel cells prepared by supercritical deposition technique, Journal of Power Sources, Volume 179, Issue 2, 2008, Pages 532-540, ISSN 0378-7753, https://doi.org/10.1016/j.jpowsour.2007.12.086.

Oki Sekizawa, Tomoya Uruga, Kotaro Higashi, Takuma Kaneko, Yusuke Yoshida, Tomohiro Sakata, and Yasuhiro Iwasawa. Simultaneous Operando Time-Resolved XAFS–XRD Measurements of a Pt/C Cathode Catalyst in Polymer Electrolyte Fuel Cell under Transient Potential Operations. ACS Sustainable Chemistry & Engineering 2017 5 (5), 3631-3636 DOI: 10.1021/acssuschemeng.7b0005

Taekeun Kim, Branko N. Popov, Development of highly-active and stable Pt/C catalyst for polymer electrolyte membrane fuel cells under simulated start-up/shut-down cycling, International Journal of Hydrogen Energy, Volume 41, Issue 3, 2016, Pages 1828-1836, ISSN 0360-3199, https://doi.org/10.1016/j.ijhydene.2015.11.107.

  1. Electronic conductivity plays an important role in the electrocatalyst materials. The authors need to provide the conductivity of the synthesized material samples.

Data was included into the publication.

  1. Some references such as https://doi.org/10.1039/D0CY02056G; https://doi.org/10.1039/D2DT01268E  should be mentioned when discussing the Oxygen Reduction Reaction.

Reviewer 4 Report

The manuscript concerns nitrogen-doped activated carbons as electrocatalysts for oxygen reduction reaction. Generally, N-doped carbonaceous materials are considered competitive catalyst for ORR, and indeed there are many papers concerning this issue. In fact, the number of papers devoted to N-doped carbons as ORR electrocatalysts is truly massive and there are also hundreds of review papers dealing with this topic.  It is well-established now that the N-doped carbon based ORR catalysts perform much better in pH 13 than in pH 1 and in alkaline conditions they indeed can outperform Pt, but in acidic conditions they are much, much worse than Pt. There is an important question if the materials you have prepared are truly metal-free or metal-contaminated? Is the pyridinic N the center of O2 reduction? Generally, I am very skeptical about the so-called metal-free N-doped carbo-catalysts. I think that in the introduction the authors should mention that the research concerning N-doped carbocatalysts is still poorly understood and there are many drawbacks of such catalysts, please see for instance: Transition metal impurities in carbon-based materials: Pitfalls, artifacts and deleterious effects, Carbon 168 (2020) 748, and also: Will Any Crap We Put into Graphene Increase Its Electrocatalytic Effect? ACS Nano 14 (2020) 21.

The paper is of average scientific value, it does not bring much new insight into the field of metal-free N-doped carbon electrocatalysts. Some improvements are necessary:  

In abstract: the authors say that the catalysts are evaluated as oxygen reduction catalysts in fuel cells. And then also in line 92 they authors say that: The resulting carbon materials with different pore size distributions were doped with nitrogen and tested as oxygen reduction catalysts in fuel cells. But this is not true, the catalysts were not tested in working fuel cell; to do that you would have to build MEA and construct single-cell fuel cell.

‘in oxygen reduction fuel cells’ – that is erroneous tern, we define fuel cells rather by the type of conducting membrane (PEM FC, phosphoric acid FC, AEM FC, etc…)

“making them an effective chemical activation precursor” – I do not understand this expression

This part:

the textural and electrocatalytic properties of activated carbon materials obtained from wood, water-insoluble lignocellulose pyrolysis tar, and their mixture (1:1) were studied. The samples were carbonized thermally (pyrolysis 500°C) and hydrothermally (250°C) and then chemically activated with NaOH (alkali to precursor ratio 3:1, temperature 800°C) and demineralized;

should be included in the section 2. Materials and Methods, 2.1. Nitrogen doping; otherwise this manuscript is difficult to follow. Some of the carbons were obtained from waste – what kind of waste, more details concerning the carbon preparation is necessary. Maybe the waste introduced transition metals into the carbons.

All these following symbols must be defined and decoded: AHTC-W-N, AHTC-H-N, AHTC-T-N, APYR- W- N, APYR-H-N, APYR-T-N in the 2. Materials and Methods, 2.1. Nitrogen doping, otherwise this manuscript is impossible to follow.

The yield of some materials is as low as 7% - that is very low indeed – could the authors comment on this issue.

The authors claim that from experiments and theoretical calculations described in the literature it has been shown that pyridinic-N is the most active in the oxygen reduction reaction. Well there are contradictory results concerning this issue.

It would be a good idea to differentiate that % from CHN analysis as they are wt.% but from XPS analysis the percentage is at.% - probably….

Please include and describe HR XPS spectra of C1s

It is very interesting that samples APYR-H-N and APYR-W-N exhibit very strong 2D bands in their Raman spectra. In fact, their G bands also exhibit high intensity. This indicates that these two materials indeed exhibit high degree of graphitization while other materials are highly disordered. Why APYR-H-N and APYR-W-N are highly graphitized materials? There is no explanation of this critical difference. Here, XRD analysis of the carbon samples would be very helpful to resolve this mystery.  

 Moderate editing of English language required

Author Response

There is an important question if the materials you have prepared are truly metal-free or metal-contaminated?

Thank you for the question, it is indeed important point to discuss. There is always a possibility for the metals to be leached into the material from the walls of the reactor. However, after the activation materials are always thoroughly washed with HCl and then rinsed until neutral pH. In this process carbons are not only demineralized in order to remove Na salts, but also other metals that might present. Basing on that it can be derived that metallic impurities might be found in negligible quantities embedded deeply into the particles structure.

Is the pyridinic N the center of O2 reduction? Generally, I am very skeptical about the so-called metal-free N-doped carbo-catalysts.

Nitrogen forms part of active centers on the surface of the material. There is a synergy of material morphology, functionalities and pore size distribution which ensure materials activity towards ORR.

I think that in the introduction the authors should mention that the research concerning N-doped carbocatalysts is still poorly understood and there are many drawbacks of such catalysts, please see for instance: Transition metal impurities in carbon-based materials: Pitfalls, artifacts and deleterious effects, Carbon 168 (2020) 748, and also: Will Any Crap We Put into Graphene Increase Its Electrocatalytic Effect? ACS Nano 14 (2020) 21.

Thank you, paper was altered accordingly.

The paper is of average scientific value, it does not bring much new insight into the field of metal-free N-doped carbon electrocatalysts. Some improvements are necessary:  

In abstract: the authors say that the catalysts are evaluated as oxygen reduction catalysts in fuel cells. And then also in line 92 they authors say that: The resulting carbon materials with different pore size distributions were doped with nitrogen and tested as oxygen reduction catalysts in fuel cells. But this is not true, the catalysts were not tested in working fuel cell; to do that you would have to build MEA and construct single-cell fuel cell.

Corrected for clarity.

‘in oxygen reduction fuel cells’ – that is erroneous tern, we define fuel cells rather by the type of conducting membrane (PEM FC, phosphoric acid FC, AEM FC, etc…)

Corrected for clarity.

“making them an effective chemical activation precursor” – I do not understand this expression

Corrected for clarity.

This part:

the textural and electrocatalytic properties of activated carbon materials obtained from wood, water-insoluble lignocellulose pyrolysis tar, and their mixture (1:1) were studied. The samples were carbonized thermally (pyrolysis 500°C) and hydrothermally (250°C) and then chemically activated with NaOH (alkali to precursor ratio 3:1, temperature 800°C) and demineralized;

should be included in the section 2. Materials and Methods, 2.1. Nitrogen doping; otherwise this manuscript is difficult to follow. Some of the carbons were obtained from waste – what kind of waste, more details concerning the carbon preparation is necessary. Maybe the waste introduced transition metals into the carbons.

All these following symbols must be defined and decoded: AHTC-W-N, AHTC-H-N, AHTC-T-N, APYR- W- N, APYR-H-N, APYR-T-N in the 2. Materials and Methods, 2.1. Nitrogen doping, otherwise this manuscript is impossible to follow.

Thank you, all the corrections have been made.

The yield of some materials is as low as 7% - that is very low indeed – could the authors comment on this issue.

In fact, this is not far off from the expected values – usually wood carbonization yields is 30%, and subsequent activation also 30% from that. Thus 7-15% yield range on the raw wood is not out of ordinary depending on the type of treatment. Since hydrothermally treated wood is less dense and not as condensed as pyrolyzed wood it is more susceptible to the aggressive activation procedure. This is also indirectly supported by sorption data, since all of the hydrothermally treated wood samples are considerably more mesoporous.

The authors claim that from experiments and theoretical calculations described in the literature it has been shown that pyridinic-N is the most active in the oxygen reduction reaction. Well there are contradictory results concerning this issue.

This is correct, however this is the most widespread hypothesis at the current moment, alongside with graphitic and quaternary N. Since our materials are mostly amorphous, pyridinic N is the most likely source of active centers. Hopefully more definitive studies will be available in the future.

It would be a good idea to differentiate that % from CHN analysis as they are wt.% but from XPS analysis the percentage is at.% - probably….

Please include and describe HR XPS spectra of C1s-

Spectra was included into paper.

It is very interesting that samples APYR-H-N and APYR-W-N exhibit very strong 2D bands in their Raman spectra. In fact, their G bands also exhibit high intensity. This indicates that these two materials indeed exhibit high degree of graphitization while other materials are highly disordered. Why APYR-H-N and APYR-W-N are highly graphitized materials? There is no explanation of this critical difference. Here, XRD analysis of the carbon samples would be very helpful to resolve this mystery. 

XRD study was included into paper.

Reviewer 5 Report

Manuscript ID: materials-2427639

Title: Effect of Pretreatment On The Nitrogen Doped Activated Carbon Materials Activity Towards Oxygen Reduction Reaction

Authors:
Galina Dobele, Ance Plavniece*, Aleksandrs Volperts, Aivars Zhurinsh, Daina Upskuviene, Aldona Balciunaite, Vitalija Jasulaitiene, Gediminas Niaura, Martynas Talaikis, Loreta Tamasauskaite-Tamasiunaite*, Eugenijus Norkus, Jannicke Kvello, Luis César Colmenares-Rausseo

This manuscript deals with the oxygen reduction reaction (ORR) activity of the thermally and hydrothermally reduced carbon materials derived from lignocellulose pyrolysis tar. The authors have used a two-step process with a first step based on either pyrolysis or hydrothermal reduction and a second step of NaOH activation, after which the samples are doped with nitrogen and studied for their activity towards the ORR in alkaline media. Overall, the experimental work is comprehensive and the authors present compelling reasons why their catalysts might be advantageous to commercial Pt/C. There are some minor issues which require additional discussion. Specific issues:

Comments to the Authors:

1)      There is no information in the manuscript at all on what is the difference between the precursors or synthesis parameters of the samples. The authors have referenced to their earlier publication describing this but a short description should at least be included in the supplementary information.

2)      The authors write in the results and discussion section that their samples were also activated using NaOH, but there is no mention of this in the experimental section. This step should also be detailed.

3)      The authors present no data on the ash content of their materials. Since metal additives can have a profound effect on the ORR activity of catalyst materials, the elemental composition (other than CHNSO) should also be perhaps presented.

4)      The RDE curves have no defined diffusion limited current plateau, which can make the analysis of the data quite difficult. How were the jd values derived in this case?

Minor remarks:

1)      Reference 44 should be corrected.

Author Response

  • There is no information in the manuscript at all on what is the difference between the precursors or synthesis parameters of the samples. The authors have referenced to their earlier publication describing this but a short description should at least be included in the supplementary information.

The paper was improved accordingly.

  • The authors write in the results and discussion section that their samples were also activated using NaOH, but there is no mention of this in the experimental section. This step should also be detailed.

The paper was improved accordingly.

3)      The authors present no data on the ash content of their materials. Since metal additives can have a profound effect on the ORR activity of catalyst materials, the elemental composition (other than CHNSO) should also be perhaps presented.

To rectify this issue XRD data were added into the paper.

4)      The RDE curves have no defined diffusion limited current plateau, which can make the analysis of the data quite difficult. How were the jd values derived in this case?

The jd values were derived at 0.25 V.

Round 2

Reviewer 4 Report

The authors corrected the manuscript significantly and they took all the comments into account. They put a lot of work and effort to improve the paper. Now the paper is much better and it presents much higher scientific values. I am glad that the authors added XRD analysis but please add information concerning the X-ray source – Cu-K-alpha probably? It is interesting that in the XRD patterns the authors managed to differentiate the 100 and 101 reflections. This is surprising because 101 reflection appears ONLY for graphite and the question is: are the materials really that highly graphitized, and if so – why? The authors detected some iron oxides in the materials (i.e., some reflections assigned to iron oxides). Please be aware that the 101 reflection of graphite overlaps with reflection from metallic iron – and then maybe the 101 is really iron impurity... Having said that, in the discussion or in the introduction or even conclusion for that matter, the authors should add comments on the problems concerning metal-free carbocatalysis. It is of great importance to realize that the research concerning metal-free nitrogen-doped carbon catalysts is still very poorly understood and there are many unknowns concerning such catalysts, please see for instance: Transition metal impurities in carbon-based materials: Pitfalls, artifacts and deleterious effects, Carbon 168 (2020) 748, and also: Will Any Crap We Put into Graphene Increase Its Electrocatalytic Effect? ACS Nano 14 (2020) 21. I still encourage the authors to consider this problem for the sake of science integrity but the final decision is theirs. Now it is even more important as the peak observed at 37.5° was allocated as magnetite (Fe3O4) in the structure of the studied materials. Metal impurities may appear in the pyrolysis process – indeed and they may hold on to the carbon matrix.

Author Response

Thank you, we took your suggestions into account and made some minor corrections.